# Molecular Physiological Characterization of a High Heat Resistant Spore Forming *Bacillus subtilis* Food Isolate

**DOI:** 10.3390/microorganisms9030667

**Published:** 2021-03-23

**Authors:** Zhiwei Tu, Peter Setlow, Stanley Brul, Gertjan Kramer

**Affiliations:** 1Laboratory for Molecular Biology and Microbial Food Safety, University of Amsterdam, 1098 XH Amsterdam, The Netherlands; Z.Tu@uva.nl; 2Laboratory for Mass Spectrometry of Biomolecules, University of Amsterdam, 1098 XH Amsterdam, The Netherlands; g.kramer@uva.nl; 3Department of Molecular Biology and Biophysics, UCONN Health, Farmington, CT 06030-3303, USA; setlow@nso2.uchc.edu

**Keywords:** *Bacillus subtilis* A163, proteome, high heat resistance

## Abstract

Bacterial endospores (spores) are among the most resistant living forms on earth. Spores of *Bacillus subtilis* A163 show extremely high resistance to wet heat compared to spores of laboratory strains. In this study, we found that spores of *B. subtilis* A163 were indeed very wet heat resistant and released dipicolinic acid (DPA) very slowly during heat treatment. We also determined the proteome of vegetative cells and spores of *B. subtilis* A163 and the differences in these proteomes from those of the laboratory strain PY79, spores of which are much less heat resistant. This proteomic characterization identified 2011 proteins in spores and 1901 proteins in vegetative cells of *B. subtilis* A163. Surprisingly, spore morphogenic protein SpoVM had no homologs in *B. subtilis* A163. Comparing protein expression between these two strains uncovered 108 proteins that were differentially present in spores and 93 proteins differentially present in cells. In addition, five of the seven proteins on an operon in strain A163, which is thought to be primarily responsible for this strain’s spores high heat resistance, were also identified. These findings reveal proteomic differences of the two strains exhibiting different resistance to heat and form a basis for further mechanistic analysis of the high heat resistance of *B. subtilis* A163 spores.

## 1. Introduction

Strains from *Bacillus subtilis* are causative agents of food spoilage, which can cause problems in the food industry [1,2,3]. This is largely due to the high stress resistance of the spores produced by this species. In response to nutritional and environmental stresses, vegetative cells of *B. subtilis* can form metabolically dormant spores which exhibit extreme resistance properties compared to their corresponding vegetative cells [4]. Surviving spores can, depending on the environmental conditions, quickly germinate, resume vegetative growth and subsequently cause food spoilage [5]. Compared to vegetative cells, spores are protected by multiple proteinaceous coat layers [6]. In particular, the water content within the spores’ core is low, and DNA in the core is surrounded by dipicolinic acid (DPA) and saturated with protective α/β-type small acid-soluble proteins (SASPs). Many factors can affect spores’ wet heat resistance. For example, spores prepared on solid media have higher wet heat resistance than those prepared in liquid [7]. Higher sporulation temperatures can also result in higher heat resistance of spores [8]. In addition, the protein composition and DPA levels of spores are affected by sporulation conditions and temperatures [8,9,10]. With respect to heat resistance of spores of different *Bacillus* strains, two distinct groups have been identified [11]. One group, which includes *B. subtilis* strains 168 and PY79, commonly used strains in laboratory research, has spores with low heat resistance, while the other group, which includes the foodborne isolate *B. subtilis* strain A163, has spores with high heat resistance. A mobile genetic element, Tn1546, is commonly present in the high resistance strains and this transposon contains the *spoVA*^2 mob^ operon which can profoundly heighten resistance of spores to heat and pressure, encoding four genes of unknown function and a three gene *spoVA* operon, which encodes proteins involved in DPA uptake into developing spores, and its release in spore germination [12,13,14,15]. Regrettably, the proteome of strain *B. subtilis* A163 has not yet been studied extensively using high resolution mass spectrometry-based proteomics so the protein expression profile has not been compared with that of low resistance spores.

First, we confirmed the higher thermal resistance of *B. subtilis* A163 spores compared to spores of *B. subtilis* PY79, and that rates of DPA release at elevated temperatures were much slower from A163 spores than from PY79 spores. We also determined the proteomes of spores and cells of *B. subtilis* A163. This work found 2011 spore and 1901 cell proteins, while 2045 cell and 2170 spore proteins were identified in *B. subtilis* strain PY79, a prototrophic derivative of strain 168 [16]. Among the proteome of spores of *B. subtilis* A163 were five of the seven proteins of the *spoVA*^2mob^ operon found, and homologs of SpoVM, which are important in spore coat assembly in PY79 [17] were not found. Previous studies show that strain A163 and strain 168 are very similar at the genomic level [18]. By searching every protein sequence of *B. subtilis* PY79 against all protein sequences of *B. subtilis* A163, 4030 protein sequences of two strains showed more than 50% identity. Among them, 1312 and 1276 proteins were quantified between the two strains in spores and cells, respectively. Both high-abundance and low-abundance proteins were revealed in spores and cells. In spores of *B. subtilis* A163, the high-abundance proteins are enriched in the Uniprot categories glycosyltransferases and proteases, while low-abundance proteins are mainly enriched in membrane and sporulation proteins. In cells of *B. subtilis* A163, high-abundance proteins are enriched in the transport of proteins and peptides, as well as competence. Proteins involved in biosynthesis and metabolism of fatty acids and lipids, as well as oxidoreductase, are enriched in the low-abundant cellular proteins.

## 2. Materials and Methods

### 2.1. Strains and Sporulation

Low spore wet heat resistant *B. subtilis* strain PY79 and the high heat resistant spore forming food isolate *B. subtilis* A163 were used in this study [19]. Both strains were sporulated in shake flasks containing 3-(N-morpholino) propane sulfonic acid (MOPS) (Sigma-Aldrich, St. Louis, MI, USA) buffered defined liquid medium [20]. In brief, a single colony from a Lysogeny broth (LB) [21] agar plate was inoculated in 5 mL LB liquid medium and cultivated until its exponential phase at 37 ℃ and 200 rpm. The exponentially growing cells were then subjected to overnight growth in MOPS medium in a series of dilutions. The dilution with exponential growth was selected and 1 mL was transferred into 100 mL of MOPS medium and allowed to sporulate for 72 h. Spores were harvested and purified with Histodenz (Sigma-Aldrich, St. Louis, MI, USA) gradient centrifugation [22]. Vegetative cells were harvested from LB medium in the exponential phase. Three biological replicates of each strain were harvested for both spores and cells and stored at −80 ℃ for further experiments.

### 2.2. Heat Resistance and DPA Measurement in Spores

Heat resistance of spores was tested at 85 ℃ and 98 ℃ following an established protocol [9]. One ml of spores with an OD_600_ of 2 (~2 × 10^8^ spores/mL) was heat activated at 70 ℃ for 30 min in a water bath. After being placed on ice for 15 min, spores were injected into a metal screwcap tube with 9 mL sterile milli-Q water pre-warmed for 20 min in a glycerol bath (85 ℃ or 98 ℃). The metal tube was then kept at 85 ℃ (or 98 ℃) for another 10 min, after which, the tube was cooled on ice. The fraction of surviving spores after heat treatment was estimated by counting the number of colonies formed on LB plates. Three biological replicates were performed for each strain tested at both 85 ℃ and 98 ℃.

Spore DPA content was calculated as μg of DPA per mg dry weight of spores. The protocol of DPA measurement was modified from [23]. One ml of spores with an OD_600_ of 2 from each strain was suspended in a buffer containing 0.3 mM (NH_4_)_2_SO_4_, 6.6 mM KH_2_PO_4_, 15 mM NaCl, 59.5 mM NaHCO_3_ and 35.2 mM Na_2_HPO_4_. For total DPA measurement, the suspended spores were autoclaved at 121 ℃ for 15 min. After incubation, the sample was centrifuged at 15,000 rpm for 2 min and 10 µL supernatant was added to 115 µL of buffer 2 (1 mM Tris, 150 mM NaCl) with and without 0.8 mM terbium chloride (Sigma-Aldrich, St. Louis, MI, USA). After 15 min of incubation, the fluorescence of the samples was measured using a Synergy Mx microplate reader (BioTek; 270-nm excitation; 545-nm reading; gain, 100) (Bad Friedrichshall, Germany). The background fluorescence (without terbium, incubated at 37 ℃) was subtracted from that of all the samples. A calibration curve of 0–125 µg/mL DPA (2,6-pyridinedicarboxylic acid) (Sigma-Aldrich, St. Louis, MI, USA) was used to calculate DPA concentrations of the sample. Samples incubated at 98 ℃ for 1–6 h were also used to measure the amount of DPA released at various heating times. Spore dry weights were determined by weighing overnight freeze-dried spores. Three biological replicates were measured for all conditions.

### 2.3. Proteome Databases and Comparison of Protein Sequences

Amino acid sequences encoded in the genome of *B. subtilis* A163 were acquired from [24] with accession no. JSXS00000000. The UniProt proteome database UP000001570 was used for *B. subtilis* PY79 [25]. Every protein sequence within UP000001570 was searched against the database of *B. subtilis* A163 to find the best match(es) using NCBI BLAST+ BLASTP (Galaxy version 0.3.3) embedded in the web-based platform Galaxy Europe (https://usegalaxy.eu/, accessed at 5 January 2021) with the E-value set at 0.00001 [26,27,28]. The protein sequences encoded in the *spoVA*^2mob^ operon in *B. subtilis* B4417 were acquired from NCBI with the reference sequence NZ_LJSM01000045.1.

### 2.4. Data Acquisition for Proteomic Analysis

Processing of samples and fractionation of every trypsin digested sample into 10 fractions was done following the protocol described by Tu et al. [29]. Every fraction was reconstituted in 0.1% formic acid in water and 200 ng equivalent (set by measuring absorbance at a wavelength of 205 nm [30]) was injected by a Ultimate 3000 RSLCnano UHPLC system (Thermo Scientific, Germeringen, Germany) onto a 75 ¼m × 250 mm analytical column (C18, 1.6 ¼m particle size, Aurora, Ionopticks, Australia) kept at 50 ℃ at 400 nL/min for 15 min in 3% solvent B before being separated by a multi-step gradient (Solvent A: 0.1% formic acid in water, Solvent B: 0.1% formic acid in acetonitrile) to 5% B at 16 min, 17% B at 38 min, 25% B at 43 min, 34% B at 46 min, 99% B at 47 min held until 54 min returning to initial conditions at 55 min equilibrating until 80 min. 

Eluting peptides were sprayed by the emitter coupled to the column into a captive spray source (Bruker, Bremen, Germany) with a capillary voltage of 1.5 kV, a source gas flow of 3 L/min of pure nitrogen and a dry temperature setting of 180 ℃, attached to a timsTOF pro (Bruker, Bremen, Germany) trapped ion mobility, quadrupole, time of flight mass spectrometer. The timsTOF was operated in PASEF mode of acquisition. The TOF scan range was 100–1700 *m/z* with a tims range of 0.6–1.6 V·s/cm^2^. In PASEF mode a filter was applied to the *m/z* and ion mobility plane to select features most likely representing peptide precursors, the quad isolation width was 2 Th at 700 *m/z* and 3 Th at 800 *m/z*, and the collision energy was ramped from 20–59 eV over the tims scan range to generate fragmentation spectra. A total number of 10 PASEF MS/MS scans scheduled with a total cycle time of 1.16 s, scheduling target intensity 2 × 10^4^ and intensity threshold of 2.5 × 10^3^ and a charge state range of 0–5 were used. Active exclusion was on (release after 0.4 min), reconsidering precursors if ratio current/previous intensity >4.

### 2.5. Data Processing

Generated data for spores (of two strains) and cells (of two strains) were processed with MaxQuant (Version 1.6.14, Martinsried, Germany) in two separate analyses [31]. 10 fractions from the same sample were set as one experiment. Proteome databases for *B. subtilis* A163 and *B. subtilis* PY79 were included in the analysis. The proteolytic enzyme used was trypsin/p, and the maximum missed cleavages was set to 2. Carbamidomethyl (C) was set as fixed modification with variable modifications of Oxidation (M) and Acetyl (Protein N-term). The type of Group specific parameters was set as TIMS-DDA. The rest of the parameters were set using the default. Since two databases were used in the analysis, the quantified values of proteins from two strains with high percentage of identity were sometimes reported as two separate proteins in the output proteinGroup.txt (vide infra). To quantitatively compare the homologous protein from two strains, we re-assembled the identified peptides in the evidence.txt file and quantified the protein amounts using the R software package *iq* [32]; the R-script and evidence.txt files used can be found in Appendix A. The minimum number of peptides for the quantification was 2. The differentially presented proteins in cells and spores were determined by using R/Bioconductor software package *limma* [33]. DAVID Bioinformatics Resources tool (version 6.8) was used to retrieve the UniProt keyword enrichment of the differentially presented proteins [34,35]. The protein list of coat proteins was retrieved from *Subti*Wiki (http://subtiwiki.uni-goettingen.de/, accessed at 5 January 2021) [36].

## 3. Results and Discussion

### 3.1. Heat Resistance and DPA Measurement of B. subtilis Spores

Spores of *B. subtilis* A163 are reported to show extreme wet heat resistance [11,18,19]. To confirm this, spores of *B. subtilis* A163 and PY79 were heat treated at 85 ℃ and 98 ℃, and as expected there was a significant decrease in surviving PY79 spores treated at 98 ℃, but no such difference for A163 spores (Figure 1A). Our measurements also showed that while *B. subtilis* A163 spores gave slightly higher values for DPA than PY79 spores, the difference was not statistically significant (Figure 1B). This latter finding is consistent with the essentially identical DPA levels found recently in *B. subtilis* with and without transposon Tn1546 [15]. However, the rate of release of DPA during heat treatment of A163 spores was much slower than from PY79 spores (Figure 1C). Treatment of spores of PY79 spores at 98 ℃ resulted in release of ~80% DPA at 1 h and almost all DPA at 5 h. However, only trace amounts of DPA were released from A163 spores during the first hour of heat treatment and only ~20% of the DPA was released at 6 h. This finding confirms a previous report [19], that A163 spores need much higher temperatures to release their DPA than low wet heat resistance spores. Killing of low wet heat resistance spores by wet heat appears due to damage to key spore proteins, and this takes place before loss of DPA presumably due to spore inner membrane (IM) damage but prevents the outgrowth of spores [37,38]. However, the high wet heat resistant A163 spores have features to maintain the native state of key spore proteins as well is the IM impermeability during heat treatment at higher temperatures.

### 3.2. Identification and Quantification of Proteomes of Spores and Cells

One important field in the study of spore forming bacteria is identification and quantification of the spore and growing cell proteome. In this study, we investigated the proteome of both spores and cells of *B. subtilis* A163. 2011 and 1901 proteins were separately identified in at least two of three biological replicates of spores and cells of *B. subtilis* A163, while with *B. subtilis* PY79, 2170 spore proteins and 2045 cellular proteins were identified. Lists of identified proteins in spores and cells can be found in Appendix A. In terms of identification of proteins from the *spoVA*^2mob^ operon in *B. subtilis* A163 spores, five proteins were identified (Table 1), excluding the protein with both a predicted DUF 421 domain and a DUF 1657 domain (2Duf protein), which is thought to be the most important one in the *spoVA*^2mob^ operon [12], as well as the SpoVAEb protein. In addition, homologs for the two DUF 1657 domain-containing proteins and SpoVAC^2mob^ were also identified. This could be because multiple copies of *spoVA*^2mob^ were present in *B. subtilis* A163 [12]. To compare proteomes of spores or cells of two strains, we first checked how much similarity there was between the protein sequences of the two strains. Among 4800 protein coding genes of *B. subtilis* A163, amino acid sequences of 4141 genes show a minimum 22% of identity with the lab strain, and 4030 genes show more than 50% identity (Figure 2). Quite a number of proteins from the two strains show a high percentage of identity, indicating that these proteins could be considered homologs. Coat proteins identified in *B. subtilis* spores are shown in Table 2. SpoVM, CotU, CotR and YjdH have no homologous proteins found in *B. subtilis* A163. SpoVM is a key protein for the proper assembly of the spore coat [17]. Two homologous genes were found for *oxdD, yjqC* and *cotF* in *B. subtilis* A163, but only one homolog of CotF and two homologs of YjqC were identified. For the proteins involved in germination and the endogenous SpoVA channel [5], some germinant receptor proteins, most notably GerB proteins, were only identified in *B. subtilis* PY79, but not in *B. subtilis* A163 (Table 3). Moreover, strain specific proteins were identified in both strains. Among the *B. subtilis* A163 specific spore proteins, none of them show predicted functions except an alpha-glucosidase (protein id = KIL30593.1).

Since two databases were used in this study, homologous proteins from the two strains were often reported as two separate results, for example RsmE (ribosomal RNA small subunit methyltransferase E, Uniprot ID P54461) identified in the cellular proteome. In a comparison of amino acid sequences, RsmE from the two strains had more than 99% identity with one amino acid difference, T18A (Figure 3A). However, two proteins were in the output as two items with their own quantitative values (Figure 3B). By checking their peptide composition, we found they both contain shared peptides which can be identified from either of the two proteins, and specific peptides caused by the T18A change. Quantification of RsmE using either of the two outputs may result in an incorrect conclusion. To overcome this issue, we re-assembled the identified peptides to include both the shared peptides and specific peptides and calculated the protein abundances accordingly [32]. In the new output, proteins from two strains having shared peptides were treated as homologous proteins for the moment and their protein identifiers were both shown in the column of Protein IDs (such as RsmE in Figure 3C). In total, 1312 and 1276 proteins were quantified between two strains in spores and cells, respectively, with at least two quantified values in each strain (Appendix A). 

Among the proteins quantitated, proteins between two strains with an identity higher than 50% were subjected to further analysis. Some proteins have multiple homologs identified in *B. subtilis* A163, such as YjqC, but only one homolog in our data is quantitively compared with the protein in *B. subtilis* PY79. That is because in a comparison of one protein between two samples, the quantification algorithm requires that some peptides identified in one sample must be also identified in the other sample [39]. In our data, since not enough peptides were identified for the other homologs, this makes them impossible to quantitively analyze. In the quantified spore proteins, 39 proteins were found to be highly abundant in *B. subtilis* A163, while 69 were low abundance (Figure 4A). For the analysis of cellular proteins, 32 and 61 proteins in *B. subtilis* A163 were present at high and low abundance, respectively (Figure 4A). In the known spore coat proteins retrieved from SubtiWiki [36], YmaG and CgeA were low abundance and CotJC, CotH, CotSA, SpoVID and GerT were high abundance in spores of *B. subtilis* A163. CgeA is a protein located in the spore crust, the outermost layer, and is considered to play a role in crust glycosylation [40]. SpoVID and CotH are essential for spore coat morphogenesis, and a *spoVID* mutant fails to encase the spore inner and outer coat layers [41,42]. *cotH* mutant spores have normal heat resistance but are deficient in several coat proteins [43]. CotJC upregulation was observed in spores from a sporulation that was *kinA*-induced, and these spores had higher wet heat resistance than when sporulation was induced by nutrient depletion [29]. However, how increased levels of SpoVID, CotH and CotJC affect spore resistance is not known. GerT is also a component of the spore coat and Δ*gerT* spores respond poorly to multiple germinants [44]. For the small acid soluble proteins and the proteins involved in spore germination (germinant receptors, SpoVA channel proteins, SleB and CwlJ) [5], none were quantified to be more or less abundant in the two strains analyzed (Appendix A). For proteins encoded in the *spoVA*^2mob^ operon, none of them are present in PY79 strain and thus were not quantitatively compared with any proteins identified in PY79 strain.

The Uniprot terms enriched from the most differentially presented spore and cellular proteins are shown in Figure 4B. Glycosyltransferases and proteases are enriched in the high abundance spore proteins of *B. subtilis* A163. Of the glycosyltransferases, YtcC is a product of the *ytcABC* operon, which could be involved in the extensive glycosylation of the spore surface [45]. YdhE plays roles in the resistance to bacterial toxins [46]. The pyrimidine biosynthetic (*pyr*) gene cluster includes the gene for PyrE [47], one of the high abundance glycosyltransferases. The last high abundance glycosyltransferase is the coat protein CotSA [48]. Among the proteases, IspA is an intracellular serine protease, and an *ispA* null mutant showed a decreased sporulation in at least one medium [49]. While AprE is one of the major extracellular alkaline proteases [50], serine protease YtrC has been reported to be in the spore IM fraction and to play a pivotal role in spore germination [51,52]. Proteins enriched in Uniprot terms sporulation and (cell) membrane are the major group among the low abundance spore proteins of *B. subtilis* A163. The proteins enriched in sporulation are listed in Table 4. Their contribution to spore resistance is unknown. Among them, SpoIIIAG, YabP, OppA, OppB, OppC, OppF, DppE and PbpE are also membrane proteins. High abundance proteins in cells of *B. subtilis* A163 are enriched in the transport of proteins and peptides, as well as competence. Proteins involved in biosynthesis and metabolism of fatty acids and lipids, as well as oxidoreductase, are enriched in the low-abundance cellular proteins of *B. subtilis* A163.

Multiple factors can contribute to the wet heat resistance of spores [65]. Among them, a *spoVA*^2mob^ operon is considered to play roles in elevation of their resistance to heat and pressure. Measurement of the DPA content of spores of *B. subtilis* A163, the parental strain containing the *spoVA*^2mob^ operon, indicated that these spores do not contain statistically significantly higher levels of DPA than those of a low resistance strain, and recent work has also found that the core water content of spores with the *spoVA*^2mob^ operon is identical in a strain lacking this operon [15]. Of the seven proteins encoded in the *spoVA*^2mob^ operon, we identified five in A163 spores, but none were the 2Duf protein thought to be of most importance in these spores high heat resistance or SpoVAEb. Furthermore, a number of germinant receptors were not identified in spores of *B. subtilis* A163. A practical approach might be trying to focus on identification of the proteome of the spore IM, as this analysis has been done on *B. subtilis* strain 1A700 [51]. On the other hand, the rate of DPA release from spores of *B. subtilis* A163 during heat treatment is low, although the mechanism preventing faster DPA release during heat treatment is unknown. Presumably this is due to the integrity and impermeability of the IM and protection by coat layers. In addition, the A163 spore proteome contains proteins specific to this strain, but with unknown function. The location of these proteins in spores and their contribution to spore resistance are also unknown. In addition, some proteins have multiple homologs identified in *B. subtilis* A163, for example coat protein YiqC. However, no homologs were found in *B. subtilis* A163 for proteins important for spore morphogenesis, such as SpoVM. What proteins would supplement the function of SpoVM or how the spore completes the coat encasement without SpoVM homologs would certainly be worth investigating. In addition, high and low abundant A163 spore proteins were revealed for the coat layers and a number of the Uniprot categories for both cellular and spore proteins. Those could also play some role in the observed high thermal resistance of A163 spores.

## 4. Conclusions

*B. subtilis* A163 arouses the interest of researchers due to its ability to produce high heat resistant spores. In this study, we found the release of spore DPA from *B. subtilis* A163 at 98 ℃ was much slower than that from *B. subtilis* PY79 spores. How the spore prevented the DPA from rapidly being released during heat treatment and if this is related with the high heat resistance of A163 spores is unknown. Through the extensive study of the proteomes of *B. subtilis* A163 and PY79 spores and cells, the proteomic differences of the two strains are revealed. This provides novel insights on the putative molecular basis of spore high wet heat resistance. Open questions include whether 2Duf is really not expressed during the *B. subtilis* A163 life-cycle, and more generally, how the proteins encoded by the *spoVA*^2mob^ operon contribute to the observed high wet heat resistance of *B. subtilis* A163 spores.

## Figures and Tables

**Figure 1 microorganisms-09-00667-f001:**
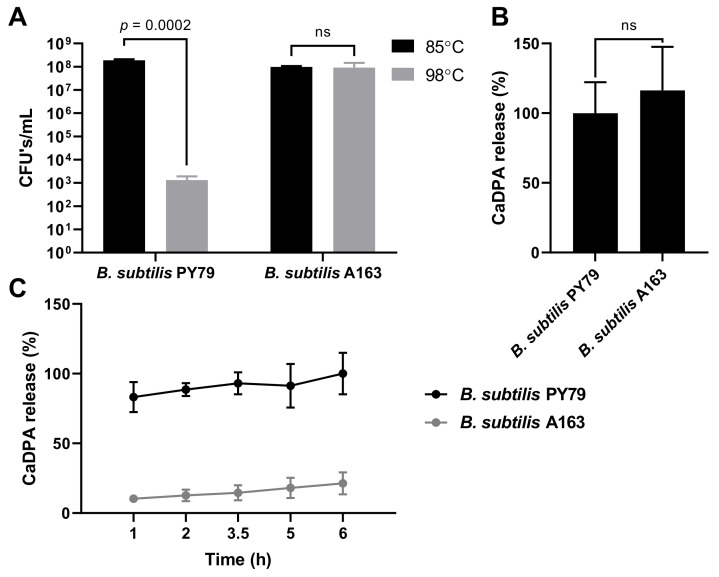
Heat resistance test and DPA content of *B. subtilis* spores. The standard deviation is shown in the graphs. Statistical significance was determined using Student’s *t*-test. ns, not significant. (**A**) Numbers of colonies formed on LB agar plates of *B. subtilis* spores wet heat treated at 85 ℃ and 98 ℃ for 10 min. (**B**) Amount of CaDPA released by *B. subtilis* spores that were autoclaved at 121 ℃. The amount of CaDPA released by spores was calculated as % of CaDPA released by *B. subtilis* PY79. (**C**) CaDPA released by *B. subtilis* spores heat treated at 98 ℃ for 1–6 h. The amount of CaDPA released by spores was calculated as % of the total CaDPA.

**Figure 2 microorganisms-09-00667-f002:**
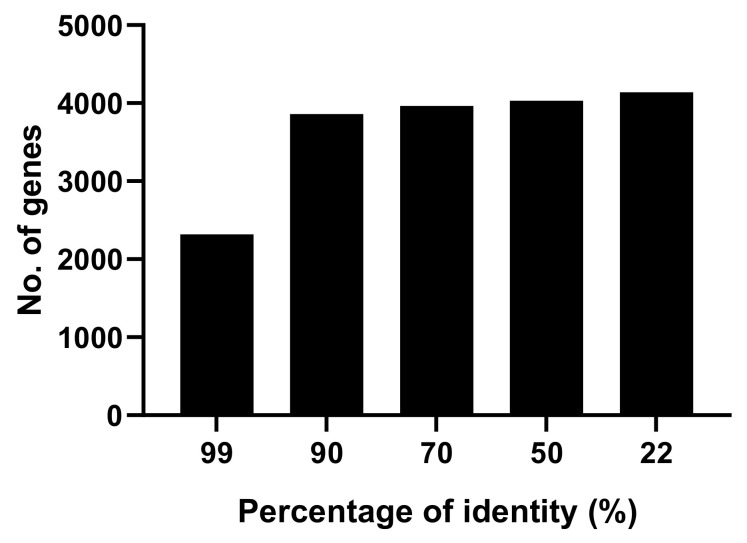
Amino acid sequence comparisons between *B. subtilis* strains. Every protein sequence in the genome of *B. subtilis* PY79 was searched against the database containing all the protein sequences of *B. subtilis* A163. The match with the highest percentage of identity was included in the figure.

**Figure 3 microorganisms-09-00667-f003:**
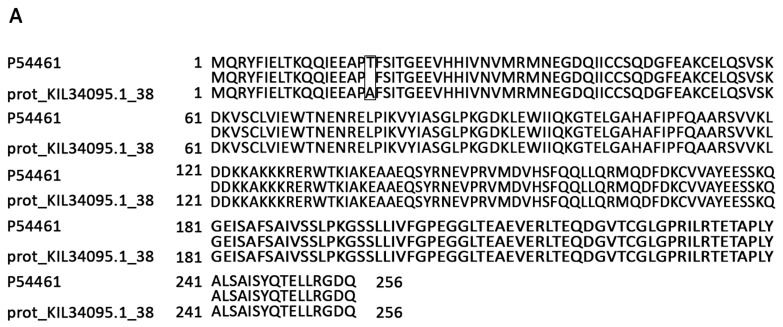
Quantitative comparison of RsmE peptide levels in growing *B. subtilis* PY79 and A163 cells. prot_KIL34095.1_38, identifier of the homologous protein of RsmE in *B. subtilis* A163; LFQ, Relative label-free quantification [39]. (**A**) Alignment of amino acids of RsmE between two strains. (**B**) Protein intensities of RsmE and their identified peptides in the default output. (**C**) Protein intensities of RsmE and their peptide components in the new output.

**Figure 4 microorganisms-09-00667-f004:**
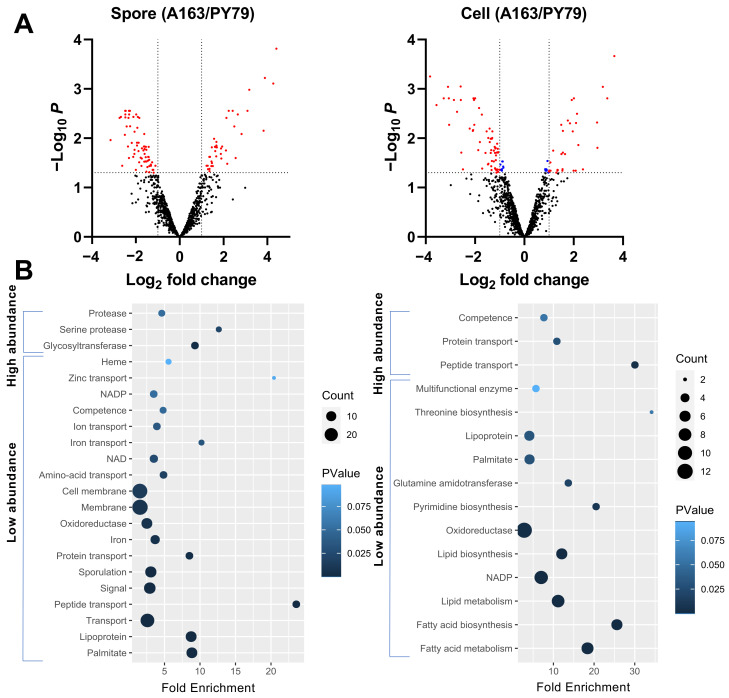
Quantitative comparison of proteomes in spores and cells of *B. subtilis* PY79 and A163. (**A**) Volcano plots of the quantified proteins in spores of the two strains and the proteome comparison of their correspondent cells. Log_2_ fold changes smaller than 0 (or larger than 0) indicate proteins with low (or high) abundance in *B. subtilis* A163. Dots in red indicate proteins in *B. subtilis* A163 that were differentially present more than twofold with *p* < 0.05. Dots in blue indicate proteins that were present with Scheme 0. but less than twofold. (**B**) Uniprot categories enrichment of the differentially presented proteins in spores and cells of *B. subtilis* A163. The fold enrichment is defined as the ratio of two proportions. The first proportion is the quantified proteins belonging to a UniProt category divided by all high- (or low-) abundant proteins. The second proportion is all proteins belonging to the UniProt category in the genome divided by the total proteins in the genome. The size of the dots is indicative of the number of quantified proteins (Count) belonging to a particular term, as shown in the legend. The color of the dots is corresponding to the Fisher exact p-value (PValue), again as shown in the legend.

**Table 1 microorganisms-09-00667-t001:** *spoVA*^2mob^ proteins identified in *B. subtilis* A163 spores.

Genes	Identifier in *B. subtilis* B4417 ^a^	Identifier in *B. subtilis* A163 ^b^	Percentage of Identity (%) ^c^	Number of Peptides
*DUF1657*	prot_WP_009336483.1_679	prot_KIL30783.1_1	100	27
		prot_KIL30530.1_3	80.882	17
*spoVAD^2mob^*	prot_WP_013352386.1_681	prot_KIL30785.1_3	100	8
*spoVAC^2mob^*	prot_WP_013352385.1_682	prot_KIL30782.1_3	100	5
		prot_KIL32093.1_7	97.183	11
*Yhcn/YlaJ*	prot_WP_017697692.1_683	prot_KIL30781.1_2	100	9
*DUF1657*	prot_WP_009336488.1_684	prot_KIL30780.1_1	100	17
		prot_KIL32090.1_4	91.176	16

^a^*B. subtilis* 168 carrying the *spoVA*^2mob^ operon [12]. ^b^ identified by searching protein sequences of *spoVA*^2mob^ operon of *B. subtilis* B4417 against the protein sequences of *B. subtilis* A163. ^c^, percentage of identity of the proteins between *B. subtilis* B4417 and *B. subtilis* A163.

**Table 2 microorganisms-09-00667-t002:** Identified coat proteins in spores of *B. subtilis* PY79 and A163.

Proteins	UniProt IDs	Homologous Proteins in *B. subtilis* A163	Percentage of Identity (%)	Number of Peptides	
				*B. subtilis* A163	*B. subtilis* PY79
SpoIVA	P35149	prot_KIL33240.1_118	99.797	131	90
SpoVID	P37963	prot_KIL31245.1_33	95.549	22	9
SpoVM	P37817	NA	NA	NA	2
YaaH	P37531	prot_KIL32101.1_8	96.721	117	132
YuzC	O32089	prot_KIL29514.1_5	96.721	13	14
CotE	P14016	prot_KIL29844.1_246	100	58	53
CotM	Q45058	prot_KIL31177.1_55	98.387	5	5
CotO	O31622	prot_KIL30325.1_87	99.111	9	8
YhjR	O07572	prot_KIL33619.1_38	100	21	24
YknT	O31700	prot_KIL30843.1_42	98.754	2	4
YncD	P94494	prot_KIL31326.1_4	99.746	40	38
CotZ	Q08312	prot_KIL30327.1_89	100	46	35
CwlJ	P42249	prot_KIL33394.1_41	95.775	36	26
YisY	O06734	prot_KIL33591.1_10	98.134	81	73
YsxE	P37964	prot_KIL31244.1_32	98.534	10	4
YutH	O32123	prot_KIL29568.1_59	99.11	16	2
CotT	P11863	prot_KIL30718.1_123	98.78	48	31
YybI	P37495	prot_KIL31595.1_71	93.893	47	51
CotA	P07788	prot_KIL30080.1_5	99.61	170	166
CotB	P07789	prot_KIL31029.1_37	68.116	4	61
CotG	P39801	prot_KIL31027.1_35	92.308	86	45
CotP	P96698	prot_KIL30034.1_53	97.203	24	14
CotQ	O06997	prot_KIL33096.1_104 (a, b)	25	0	111
CotS	P46914	prot_KIL33875.1_18	99.145	133	123
CotW	Q08310	prot_KIL30330.1_92	98.095	22	25
LipC	P42969	prot_KIL29460.1_9	100	72	55
OxdD	O34767	prot_KIL33540.1_11 (a)	94.231	0	70
		prot_KIL33541.1_12 (a)	99.642	0	
Tgl	P40746	prot_KIL32205.1_19	99.184	56	40
YjqC	O34423	prot_KIL32111.1_3 (b)	34.426	151	46
		prot_KIL30071.1_90	51.163	1	
YjzB	O34891	prot_KIL30279.1_41	96.104	1	4
YmaG	O31793	prot_KIL29859.1_261	96.703	7	19
YppG	P50835	prot_KIL33183.1_61	97.6	4	6
YtxO	P46916	prot_KIL33874.1_17	95.804	52	47
YxeE	P54944	prot_KIL32257.1_32	100	17	15
CotU	O31802	NA	NA	NA	27
CgeA	P42089	prot_KIL31959.1_13	96.241	12	8
CgeB	P42090	prot_KIL31960.1_14	94.386	11	3
CgeC	P42091	prot_KIL31958.1_12	98.02	2	3
CgeE	P42093	prot_KIL31956.1_10	99.228	14	21
CmpA	P14204	prot_KIL33555.1_2	100	2	10
CotC	P07790	prot_KIL31334.1_12	100	48	30
CotF	P23261	prot_KIL31586.1_62 (a)	100	0	51
		prot_KIL31587.1_63	94.643	3	
CotH	Q45535	prot_KIL31028.1_36	97.238	101	77
CotI	O34656	prot_KIL33877.1_20	98.3	115	107
CotJA	Q45536	prot_KIL30149.1_74	98.78	67	53
CotJB	Q45537	prot_KIL30150.1_75	100	10	12
CotJC	Q45538	prot_KIL30151.1_76	100	56	41
CotR	O06996	NA	NA	NA	70
CotSA	P46915	prot_KIL33876.1_19	99.469	185	116
CotX	Q08313	prot_KIL30329.1_91	100	72	48
CotY	Q08311	prot_KIL30328.1_90	100	50	39
GerQ	P39620	prot_KIL31830.1_12	98.895	21	17
GerT	Q7WY67	prot_KIL32533.1_42	95.541	33	7
SafA	O32062	prot_KIL32613.1_30	98.45	77	74
SpsB	P39622	prot_KIL31833.1_15	98.911	27	23
YabG	P37548	prot_KIL32152.1_41	98.276	67	54
YdhD	O05495	prot_KIL30052.1_71	99.048	65	53
YgaK	Q796Y5	prot_KIL30263.1_25 (a, b)	24.691	0	49
YhbB	O31589	prot_KIL31183.1_5	99.016	39	43
YheC	O07544	prot_KIL33710.1_129	100	48	2
YjdH	O31649	NA	NA	NA	19
YkvP	O31681	prot_KIL33950.1_72	99.248	59	49
YobN	O34363	prot_KIL30953.1_14	98.536	26	14
YodI	O34654	prot_KIL32546.1_55	95.181	10	11
YpeP	P54164	prot_KIL33154.1_32	99.558	13	17
YqfT	P54477	prot_KIL32864.1_5	100	14	18

NA, not available, no homologous proteins were found in *B. subtilis* A163 through searching using BLASTP; (a), not identified in the spore proteome, but showing a high percentage of identity with the coat protein; (b), low percentage of identity, more research is necessary.

**Table 3 microorganisms-09-00667-t003:** Identified *B. subtilis* PY79 and A163 proteins involved in germination.

Proteins	UniProt IDs	Homologous proteins in *B. subtilis* A163	Percentage of Identity (%)	Number of Peptides	
				*B. subtilis* A163	*B. subtilis* PY79
GerAA	P07868	prot_KIL32731.1_49	97.303	14	24
GerAB	P07869	prot_KIL32730.1_48 (a)	98.082	0	3
GerAC	P07870	prot_KIL32729.1_47	94.906	19	30
GerBA	P39569	prot_KIL31057.1_65 (a)	98.324	0	15
GerBC	P39571	prot_KIL31055.1_63 (a)	97.861	0	27
GerKA	P49939	prot_KIL29364.1_41	98.162	9	15
GerKB	P49940	prot_KIL29362.1_39	96.783	1	3
GerKC	P49941	prot_KIL29363.1_40 (a, b)	24.378	0	6
GerD	P16450	prot_KIL31648.1_3	100	31	52
GerPA	O06721	prot_KIL33608.1_27	98.63	6	9
GerPB	O06720	prot_KIL33609.1_28	100	6	9
GerPC	O06719	prot_KIL33610.1_29	99.024	9	3
GerPD	O06718	prot_KIL33611.1_30 (a)	100	0	1
GerPE	O06717	prot_KIL33612.1_31	99.115	5	1
GerPF	O06716	prot_KIL33613.1_32	100	3	4
CwlJ	P42249	prot_KIL33394.1_41	95.775	36	26
SleB	P50739	prot_KIL33255.1_133	90.12	46	49
SpoVAA	P40866	prot_KIL33311.1_189	99.515	1	1
SpoVAC	P40868	prot_KIL33309.1_187	99.333	13	10
SpoVAD	P40869	prot_KIL33308.1_186	99.408	94	89
SpoVAEa	P40870	prot_KIL33306.1_184	100	13	11
SpoVAF	P31845	prot_KIL33305.1_183	99.189	31	27

(a), not identified in the spore proteome, but showing a high percentage of identity with the protein; (b), low percentage of identity, more research is necessary.

**Table 4 microorganisms-09-00667-t004:** Proteins enriched in the Uniprot term of sporulation.

Proteins	Descriptions	References
SpoIIIAG	a key component of a feeding tube apparatus creating a direct conduit between the developing forespore and the mother cell	[53,54]
Spo0M	regulating progress of sporulation and expression of Spo0A, but the mechanisms is still unknow	[55,56]
SpoVIF	involved in assembly of spore coat proteins that have roles in lysozyme resistance	[57]
YabP	a coat-associated protein	[58]
SplB	UV resistance of spores, DNA repair in spore germination	[59]
SinR	the master regulator of biofilm formation	[60]
OppA, OppB, OppC, OppF, DppE	the ATP binding cassette (ABC) transporter systems	[61]
PbpE	penicillin-binding protein PBP 4	[62,63]
YraD	forespore-specific sporulation protein, similar to spore coat protein	[64]

## Data Availability

No new data were created or analyzed in this study. Data sharing is not applicable to this article.

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
