# Peer review of "Molecular Physiological Characterization of a High Heat Resistant Spore Forming Bacillus subtilis Food Isolate"

_microorganisms, 2021, doi:10.3390/microorganisms9030667_

Round 1
Reviewer 1 Report
Journal: Microorganisms
Title of the manuscript: Molecular physiological characterization of a high heat resistant spore forming Bacillus subtilis food isolate
Manuscript ID: microorganisms 1148406
In this manuscript, a study on the strain B. subtilis A163 and to its ability to produce high heat resistant spores, was carried out. In this study, the higher thermal resistance of B. subtilis A163 spores compared to spores of B. subtilis PY79 was confirmed.
Highlighting proteomic differences between the two strains, novel insights in the putative molecular basis of spore high wet heat resistance were provided. Proteomic investigation were performed, evidencing 2011 spore and 1901 cell proteins in B. subtilis A163 and 2170 spore and 2045 cell proteins in B. subtilis strain PY79 were determined.
Rates of dipicolinic acid (DPA) release at elevated temperatures were investigated, evidencing the release of spore DPA from B. subtilis A163 at 98℃ that resulted much slower than that from B. subtilis PY79 spores.
These findings reveal proteomic differences of the two strains exhibiting different resistance to heat and form a basis for further mechanistic analysis of the high heat resistance of B. subtilis A163 spores.
Investigation on how the proteins encoded by the spoVA2mob operon contribute to the observed high wet heat resistance of B. subtilis A163 spores, could be very informative.
The manuscript is interesting, well written and offers important insights and information on the high heat resistance adaptation in spores of Bacillus subtilis.
Revisions
Line 88: “2.2. Heat resistance of and DPA measurement in spores” change to “2.2. Heat resistance and DPA measurement in spores”or “2.2. Heat resistance of spores and DPA measurement”, please, check;
Line 254: Figure 3 is difficult to read, a possible change in page orientation could help;
Line 287: Figure 4 should have report clear indication about strains and spore or cells;
Line 366: Please check the Italic style of the bacterial species names in the reference list.
Author Response
Dear reviewer, hereby we upload our responses to the points put forward in your review. Our answers are in given in red. We hope that with these the revised manuscript is acceptable for publication. Kind greetings, prof. Dr. Stanley Brul (corresponding author).
Revisions
Line 88: “2.2. Heat resistance of and DPA measurement in spores” change to “2.2. Heat resistance and DPA measurement in spores”or “2.2. Heat resistance of spores and DPA measurement”, please, check;
Changed to “2.2. Heat resistance and DPA measurement in spores”
Line 254: Figure 3 is difficult to read, a possible change in page orientation could help;
We rearranged figure 3 to enhance its readability.
Line 287: Figure 4 should have report clear indication about strains and spore or cells;
In figure 4 (at line 289) we now indicate that the proteome comparison in the figure 4 is between A163 vs. PY79 spores (left hand panels) and cells (right hand panels).
Line 366: Please check the Italic style of the bacterial species names in the reference list.
Done

Reviewer 2 Report
Review of the article: “Molecular physiological characterization of a high heat resistant spore forming Bacillus subtilis food isolate”
Submission ID - microorganisms-1148406
The presented manuscript is on interesting subject – the molecular mechanisms of heat resistance of Bacillus subtilis isolates. All experiments were well planned and performed. The obtained results are interesting and the form of presentation is excellent. In my opinion the manuscript can be accepted for publication in Microorganisms. I have only several suggestions (presented below) that the authors could take into account preparing final version of the article.
Abstract – The abstract is interesting and well prepared – no critical remarks
Introduction – well prepared. I have only one comment – I would be grateful for a short information about the strains tested (B. subtlilis A163 and PY79). There is only a short information in the title that these strains are food isolates. Some more detailed information (maximum 1-2 sentences) would be interesting for readers. I understand that some references are cited, however I think that some essential information about the strains should be presented in the manuscript.
Materials and Methods
Important advantage of the manuscript is the fact that modern research techniques have been used.
Line 92 – why the metal tube was used for this experiment (just for my information I wonder if the tubes from another materials could be used)
Lines 109-110 – please explain why investigation of DPA released at various heating times is important for this study.
Results
All results are very interesting. The form of presentation of the obtained results is excellent. I was only a bit confused about results (or rather description) presented in Fig. 1B. In the legend to Fig. 1B it should be clearly written that the result presented in this figure is the amount of DPA released from the spores that were heated in autoclave (MM - line 101); alternatively it could be described as spore DPA content.
Final decision – In my opinion the manuscript can be accepted in current form.
Author Response
Dear reviewer, hereby we upload our responses to the points put forward in your review. Our answers are in given in red. We hope that with these the revised manuscript is acceptable for publication. Kind greetings, prof. Dr. Stanley Brul (corresponding author).
Revisions
Introduction – well prepared. I have only one comment – I would be grateful for a short information about the strains tested (B. subtlilis A163 and PY79). There is only a short information in the title that these strains are food isolates. Some more detailed information (maximum 1-2 sentences) would be interesting for readers. I understand that some references are cited, however I think that some essential information about the strains should be presented in the manuscript.
PY79 is a lab strain and A163 is isolated from food samples, as indicated on lines 46-47
Materials and Methods
Important advantage of the manuscript is the fact that modern research techniques have been used.
Line 92 – why the metal tube was used for this experiment (just for my information I wonder if the tubes from another materials could be used)
Metal tubes were used because metal has very good ability to conduct heat and therefore heating the sample evenly and precisely. Other materials good at heat conduction could also be used.
Lines 109-110 – please explain why investigation of DPA released at various heating times is important for this study.
From this experiment, we confirmed that A163 spores can maintain their IM impermeability to prevent the quick release of DPA at 98℃ which is a significant physiological characteristic of A163 spores which could contribute to the high thermal resistance of these spores.
Results
All results are very interesting. The form of presentation of the obtained results is excellent. I was only a bit confused about results (or rather description) presented in Fig. 1B. In the legend to Fig. 1B it should be clearly written that the result presented in this figure is the amount of DPA released from the spores that were heated in autoclave (MM - line 101); alternatively it could be described as spore DPA content.
The Legend of Figure 1B has changed to “Amount of CaDPA released by B. subtilis spores that were autoclaved at 121℃.” Line 191.
Final decision – In my opinion the manuscript can be accepted in current form.
